# Effects of Process Parameters on the Thickness Uniformity in Two-Point Incremental Forming (TPIF) with a Positive Die for an Irregular Stepped Part

**DOI:** 10.3390/ma13112634

**Published:** 2020-06-09

**Authors:** Lu Ou, Zhiguo An, Zhengyuan Gao, Shuqiang Zhou, Zhengxing Men

**Affiliations:** 1School of Mechanotronics & Vehicle Engineering, Chongqing Jiaotong University, Chongqing 400074, China; 631604011229@mails.cqjtu.edu.cn (L.O.); zhengyuangao@cqjtu.edu.cn (Z.G.); 2Mechanical and Electrical Engineering Department, Chengdu Aeronautic Polytechnic, Chengdu 610021, China; zhoushu1987qiang@163.com (S.Z.); amen1980@163.com (Z.M.)

**Keywords:** two-point incremental forming, thickness uniformity, process parameter, formability, irregular stepped part

## Abstract

Incremental sheet forming (ISF) is a novel flexible forming technology with advantages, such as a low forming force, low-energy-consuming equipment, and good forming performance. The lack of available information about the formability of the two-point incremental forming (TPIF) process makes it limited for practical applications. Taking an irregular stepped part as the target part, the effects of process parameters on the thickness uniformity when using TPIF with a positive die for AA1060 aluminum alloy sheets were investigated. First, the set of optimal parameters regarding the diameter of the tool head, feed rate, and the step size were obtained through orthogonal experiments. Furthermore, the optimal parameter set of the number of forming passes, the direction of movement of the forming tool, and the forming angle was determined and the optimal forming result was numerically and experimentally verified. This demonstrated that the parameters affecting the thickness uniformity of the irregular stepped parts were, in descending order, the diameter of the forming tool, the feed rate, and the step size, with corresponding optimal values of 12 mm, 15,000 mm/min, and 0.4 mm, respectively. With an increase of the number of passes and a decrease of the forming angle between adjacent passes, and adopting an alternating clockwise and counterclockwise toolpath, the thickness uniformity of the formed parts was effectively improved.

## 1. Introduction

The incremental sheet forming (ISF) process, as one of the research frontiers in the field of rapid sheet metal forming, can form parts with a complex curved surface shape without a special mold or using a simple mold. This process meets the requirements of small-scale and diversified economic markets worldwide, overcomes the problems of long production cycles and high costs, and has a wide range of applications from aerospace to medical research [1,2,3]. Based on the forming method, the ISF technology can be classified into single-point incremental forming (SPIF) or negative incremental forming (NIF) and two-point incremental forming (TPIF) or positive incremental forming (PIF), in which the tool moves around a partial or fully fixed die on a programmed toolpath [4,5]; schematics of ISF are shown in Figure 1.

Sheet metal forming is a nonlinear mechanical process with a large deformation and the flow of materials cannot be accurately controlled due to many factors. Thereby, there will be a variety of defects in the forming process, such as wrinkling, fracture, springback, and so on; therefore, improving formability is essential. Ren et al. [6] alleviated springback defects in sheet metal forming using a geometric complexity treatment and springback prediction. Meanwhile, Zhu et al. [7] proposed a method for predicting the springback and generating the compensated forming trajectory. Zhan et al. [8] investigated the fracture behavior during SPIF and found that the damage gradient increases gradually along the thickness direction. Wang et al. [9] proposed a process window for higher forming temperatures with a reasonable surface quality for two types of surface defects in FS-ISF, i.e., cutting and fish scale.

During the SPIF process, whether the sheet metal thickness is uniform or not directly affects the final forming quality; hence, thickness uniformity can be considered an indicator of formability to prevent the local thinning from becoming serious. Barnwal et al. [10] investigated the formability of the cone by analyzing the thickness strain distribution of the components. This study reveals that the process parameters are likely to influence the texture development, especially for large tool head diameters and vertical pitch values. Li et al. [11] studied the effects of a tool tip with a roller ball and sliding tips on the forming by evaluating the strain behavior and thickness distribution with different tools and revealed that a rolling tool tip produced better surface integrity compared with a sliding tool tip. Hamilton et al. [12] investigated the effects of feed rates in the range of 5080–8890 mm/min and tool rotating speeds in the range of 400–2000 rpm on the thickness distribution and sectional microstructure in SPIF.

The SPIF process involves several process parameters that may affect the forming behavior, and in turn, the performance and/or quality of the final product. Naranjo et al. [13] suggested that increasing the temperature could improve the forming performance of materials. Following this, some researchers proposed to increase the temperature of metal sheets through external heat sources [14,15,16], particularly for materials with poor plasticities, such as titanium and magnesium alloys. Kuma et al. [17] found the combination of higher spindle speed and sheet thickness resulted in the successful forming of components without fracture. Ghulam et al. [18] found that an increase in the wall angle, feed rate, and rotational speed could improve the microstructure, increase the strength, and reduce the ductility of aluminum alloys. Radu et al. [19] found that a large tool head diameter had positive effects on the surface roughness and microstructure of parts but negative effects on the accuracy of the formed parts. Hussain et al. [20] found that the effects of sheet thickness, wall angle, step size, and the interaction between the sheet thickness and wall angle are extremely significant for profile accuracy. Bastos et al. [21] found the feed rate has no effect on surface roughness for AA1050-H111 after the feed rate reaches up to 1500–12,000 mm/min. Golabi et al. [22] studied the impacts of the cone diameter and sheet thickness in the SPIF process on SS304 sheets. An increased sheet thickness increased the forming depth of conical frustums.

A good motion trajectory can help to improve the quality of the formed parts. Jeswiet et al. [23] found that a spiral tool path can produce a uniform strain without scratches. Lu et al. [24] proposed a bidirectional tool trajectory correction method based on model predictive control. The study results showed that in comparison with the existing forming process, the proposed forming method could effectively reduce the errors in the forming wall and base. Carette et al. [25] developed an automatic toolpath generation method based on feature geometry. Through using a path offset and compensation, an approach to optimize the surface quality was proposed and verified using simulations and experiments. Hu et al. [26] proposed a homogenization method for the sheet part thickness based on forming direction optimization. Behera et al. [27] proposed a solution to improve the accuracy by using multivariate adaptive regression splines (MARS) as an error prediction tool. Blaga et al. [28] determine the optimal forming strategy by changing the press position and path of the punch. Nirala et al. [29] compared the residual stress distribution between a fractal-geometry-based incremental toolpath (FGBIT) and conventional incremental toolpaths and found that a higher fatigue life and better strength-to-weight ratio can be formed using an FGBIT.

At present, some contributions have been made using TPIF. Siddiqi et al. [30] designed a fixture for TPIF, which can realize the function of free vertical movement while resisting horizontal distortion. Fiorentino et al. [31] studied the effects of the different tool paths on the maximum forming forces in the case of TPIF with a die and found that a low wall inclination in a positive tool path can reduce the force and improve the accuracy. Formisano et al. [32] compared the SPIF and TPIF processes based on numerical simulations and experimental evaluation; the results showed that the TPIF method can improve the formability and geometric accuracy for formed parts.

Despite the abovementioned research, the knowledge of these techniques still requires deepening, especially for TPIF. To increase available information about the formability of TPIF, an investigation of the effect of process parameters on the thickness uniformity for AA1060 aluminum alloy sheets was carried out through a combination of numerical simulations and physical experiments.

## 2. Materials and Methods

### 2.1. Materials and Geometric Model

AA1060 aluminum alloy sheets of size 200 mm × 200 mm with a thickness of 1 mm were chosen for the numerical simulations and physical experiments. The full die and blank holder were fabricated using 45 steel and a forming tool made using W18Gr4V. An irregular stepped part with an upper diameter of 67.6 mm and a vertical depth of 20 mm was designed as the target part. The 3D geometric model of the part is shown in Figure 2a and its main dimensions are illustrated in Figure 2b.

### 2.2. Methods

#### 2.2.1. Experimental Set-Up

A three-axis numerical control machining center named VM903H (SJTY, China) with Siemens 828D numerical control system was used as the forming equipment in the experiment. Figure 3 shows the forming device, which mainly consisted of a forming tool, a full die, guide elements, and a blank holder (pressing plate and pallet). The pallet could slide freely along the guide pillar, and the device was fixed on the computer numerical control (CNC) machining center. During forming, the forming tool moved to the corresponding position according to the preset numerical control program generated by the UG-NX 12.0 (Siemens PLM Software, Plano, TX, USA), and the sheet was plastically formed point by point and layer by layer. When the first layer was finished, the forming tool head was pressed down on the second layer and moved based on its motion trajectory. The process was repeated until the entire workpiece was formed. Figure 2a shows the single-pass toolpath for forming the part, where the finished product is shown in Figure 3. The formed part had an obvious crack near the protrusion due to the severe thinning of the sheet.

In this investigation, the sheet thickness was considered as an indicator to verify the correction of the numerical simulations. PRINCE 775 (SCANTECH, Hangzhou, China), a hand-held 3D laser scanner, was used for the geometric measurements of the formed parts. The light source of the scanner included 12 laser crosshairs with a measurement rate of 480,000 times per second and a maximum accuracy of 0.03 mm. The scanning process of the formed part is shown in Figure 4 and the obtained data were post-processed to compare the X-, Y-, and Z-coordinate values with those from the numerical simulation results.

#### 2.2.2. Finite Element (FE) Model

The FE commercial codes LS-DYNA were adopted for the numerical simulations. Shell elements were selected for the sheet in the forming simulations. The sheet was analyzed using the Belystchko–Wong–Chiang element algorithm. To keep an appropriate balance between the accuracy and efficiency of the calculations, a shear factor is 5/6 was used and five integration points in the thickness direction of the shell element were extracted. The sheet was considered to be a flexible body and the tool, pressing plate, and pallet were considered to be rigid bodies. Figure 5 shows the finite element mesh model. A mapped meshing was employed for the sheet metal with a mesh size of 1.5 mm. To save on calculating time, a bigger element size was taken for the blank holder and the supporting mold.

The Barlat–Lian’89 three-parameter yield model was selected and the anisotropic yield criterion under plane stress can be expressed as:(1)$$2{\left(\sigma_{\gamma }\right)}^{m}=a\left|K_{1}+K_{2}{|}^{m}+a\right|K_{1}-K_{2}{|}^{m}+c|K_{2}{|}^{m},$$
where $\sigma_{\gamma }\;$ is the yield stress, *m* is the Barlat constant (for face-centered cubic materials, *m* = 8, and for body-centered cubic materials, *m* = 6) [33], and *K*_1_ and *K*_2_ are the stress tensor invariants.
(2)$$K_{1}=\frac{\sigma_{xx}-h\sigma_{yy}}{2},K_{2}=\sqrt{{\left(\frac{\sigma_{xx}-h\sigma_{yy}}{2}\right)}^{2}+p^{2}\tau^{2}{}_{xy}},$$
(3)$$a=2-2\sqrt{\frac{R_{0}}{1+R_{0}}\times \frac{R_{90}}{1+R_{90}}},C=2-a,h=\sqrt{\frac{R_{0}}{1+R_{0}}\times \frac{1+R_{90}}{R_{90}}},$$
where *a*, *h*, and *p* are anisotropic material constants, determined by the wide-thickness strain ratio *R* values (*R*_0_, *R*_45_, *R*_90_) measured in three different directions under uniaxial stretching. For any angle φ,
(4)$$R_{\varphi }=\frac{2m\sigma_{\gamma }{}^{m}}{\left(\frac{\partial \varphi }{\partial \sigma_{xx}}+\frac{\partial \varphi }{\partial \sigma_{yy}}\right)\sigma_{\varphi }}-1,$$
where $\sigma_{\varphi }$ is the unidirectional tensile stress relative to angle  φ in the direction of rotation. The yield strength can be expressed as:(5)$$\sigma_{\gamma }=k{\left(\varepsilon_{0}+\varepsilon_{p}\right)}^{n}.$$

The yield strength is expressed in terms of the strength coefficient *k* and strain hardening exponent *n*. Because the upper pressure plate, lower pallet, tool head, and supporting mold do not yield, they were regarded as rigid bodies. Table 1 lists the material properties of the alloy used in this study.

The master–slave surface method was used for automatic contact between the sheet metal and the supporting die, where the upper and lower pressing plates had a friction coefficient of 0.2 [34]. In the forming process, the forming tool head drove the sheet metal movement according to the preset numerical control program without rotation and only linearly moved in the X- and Y-directions in each layer. The pressing plate and pallet only moved downward along the guide pillar when the forming tool pressed down. The supporting mold was stationary such that the system was restricted to a total of six degrees of freedom.

#### 2.2.3. Design of the Experiments

The maximum strain and maximum thickness difference were used to characterize the thickness uniformity. Based on the result of the numerical simulations, the strain could be read directly through the post-processing program. The thickness difference referred to the difference between the maximum and minimum wall thicknesses in the region affected by the forming process.

According to the part formed using a single-pass TPIF process in Figure 3, at least two passes were needed to form the target parts. The design of the experiments is illustrated in Figure 6. To simplify the design of the experiments, the process parameters were divided into two groups, each of which was independent and did not affect each other.

The process parameters, including the tool head diameter, feed rate, and step size, were chosen as the experimental factors for group 1 and their levels are shown in Table 2. The standard orthogonal experiment table designed using the L9 (3^3^) structure is listed in Table 3. The set of optimal process parameters was determined using a range analysis. During the forming process, two passes were used in all the experiments, both of which moved in a clockwise direction, and the forming angle of the upper part was set to 30° and the lower was set to 45° in the first pass.

Meanwhile, to present more detail regarding the relationships between the process parameters and the effective strain and thickness difference, more levels of each process parameter were needed; therefore, the intermediate values of each process parameter in group 1 were added.

Parameter group 2 covered the number of forming passes, the direction of movement of the forming tool, and the forming angle. Based on the set of optimal process parameters obtained from the above orthogonal experiment, the numerical simulations were carried out using the different process parameters in group 2.

The different numbers of forming passes, including two, three, and four passes, which affected the formability of the formed part, were taken to form the part; then, the effects of different numbers of forming passes on the thickness uniformity of the formed part was evaluated. Figure 7 shows the motion trajectory of the four passes TPIF obtained by processing the data in MATLAB R2017b (MathWorks, Natick, MA, USA); more specifically, Figure 7a–d show the motion trajectories for the first, second, third, and fourth passes, respectively.

The forming tool could be operated clockwise or counterclockwise, as shown in Figure 8. To study the effect of the movement direction of the forming tool on the uniformity of the sheet thickness, three different schemes were purposed. In Figure 9 - Scheme 1, two counterclockwise passes were set and two clockwise passes were set in Scheme 2, while in Scheme 3, the first pass was counterclockwise and the second was clockwise.

The forming angle is denoted by β in Figure 1a, where for a 1060 aluminum blank with a thickness of 1 mm, the forming limit angle was approximately 67° [35], while the maximum forming angle of the target part was 85°. Thus, the following three design schemes were proposed when the forming angle in the first pass was lower than 67° as shown in Figure 9. The target part was divided into two parts: an upper part and a lower part. The angle of the upper part and lower part are denoted by α_k_ and β_k_, respectively, where k is the experiment number, such that α_1_ > α_2_ > α_3_ > α_0_ and β_1_ < β_2_ < β_3_ < β_0_, where α_0_ and β_0_ are the last forming angles, and α_k_ and β_k_ are the forming angles in the first pass. The effect of the forming angle in the multiple passes on the sheet thickness uniformity was explored by varying the forming angle in the first pass.

## 3. Results and Discussion

### 3.1. Results of the Range Analysis

Through the range analysis, the effects of different parameters on the thickness uniformity were investigated. Table 4 lists the range analysis results of the maximum strain. Table 5 lists the range analysis results of the maximum thickness difference, where D is the range value. The factor relationship affecting the uniformity of the wall thickness of the target part was in the order A > B > C, i.e., the diameter of the tool head had the greatest effect, followed by the feed rate, and the step size has the least effect. A smaller maximum thickness difference and strain corresponded to a better uniformity and a higher forming quality. A3B1C1 represented the optimal set of process parameters: a tool head diameter of 12 mm, a feed rate of 15,000 mm/min, and a step size of 0.4 mm.

#### 3.1.1. Diameter of the Tool Head

The effect of using different tool head diameters of 8, 9, 10, 11, and 12 mm was evaluated. The relationship curves between the maximum thickness difference and the effective plastic strain as a function of the tool head diameter are shown in Figure 10a, which were fitted using polynomials to give Equations (6) and (7):(6)$$T_{d}=0.8224-0.02681D+0.000785714D^{2},$$
(7)$$S_{e}=2.90694-0.29397D+0.01039D^{2},$$
where *T_d_* is the maximum thickness difference, *D* is the diameter of the tool head, and *S_e_* is the effective plastic strain.

As the tool head diameter increased, the sheet thickness difference and effective plastic strain showed downward trends. When the diameter was 8 mm, the maximum thickness difference was 0.657 mm, and the strain value was 1.217; when the diameter was 12 mm, the maximum thickness difference was 0.615 mm and the strain value was 0.876. Therefore, we can conclude that a larger tool head diameter led to a smaller maximum thickness difference and a more uniform deformation.

#### 3.1.2. Feed Rate

Feed rates of 15,000, 16,200, 17,400, 18,600, and 19,800 mm/min were applied to form the target part, and the effects of feed rate were evaluated. The relationship curves between the maximum thickness difference and effective plastic strain as a function of the feed rate are shown in Figure 10b, which were fitted using polynomials to provide Equations (8) and (9):(8)$$T_{d}=0.78975-0.0000271429F+0.00000000109F^{2},$$
(9)$$S_{e}=5.10641-0.000575866F+0.00000002F^{2},$$
where *F* is the feed rate.

As the feed rate increased, the sheet thickness difference and the effective plastic strain showed upward trends. When the feed rate was 15,000 mm/min, the maximum thickness difference was 0.628 mm and the strain value was 0.996; when the feed rate was 19,800 mm/min, the maximum thickness difference was 0.68 mm and the strain value was 1.54. A higher feed rate led to a larger maximum thickness difference and a higher effective plastic strain, i.e., the sheet thickness became more uneven. A feed rate of 15,000 mm/min is therefore advised to ensure that the tensile stress generated during forming is low and the thickness difference of the sheet is reduced, which is beneficial for the forming.

#### 3.1.3. Step Size

Step size, denoted by ΔZ, is illustrated in Figure 1a. The values of 0.4, 0.5, 0.6, 0.7, and 0.8 mm were selected to form the stepped part, and the effects of step size were evaluated. The relationship curves between the maximum thickness difference and effective plastic strain as a function of step size are shown in Figure 10c, which were fitted using polynomials to produce Equations (10) and (11):(10)$$T_{d}=0.63154-0.01386S+0.03571S^{2},$$
(11)$$S_{e}=1.06366-0.20187S+0.25394S^{2},$$
where *S* is the feed rate.

The thickness uniformity curve shows that with an increase of step size, the thickness difference increased, i.e., the step size was proportional to the maximum wall thickness difference but inversely proportional to the thickness uniformity. When the step size was 0.8 mm, the thickness difference was the highest (0.643 mm), and the thickness dramatically decreased. When the step size was 0.4 mm, the thickness difference was the smallest (0.632 mm), and the sheet thickness uniformity was the highest. The effective plastic strain curve shows that the strain was the lowest at a step size of 0.5 mm. Compared to a step size of 0.4 mm, the difference was small and the curve exhibited an upward trend. The results of the thickness difference curve show that the smaller the step size, the better the forming quality of the part.

### 3.2. Number of Forming Passes

Figure 11 shows the thickness and effective plastic strain contours of each pass in the four-pass TPIF. In each pass, as the deformation amount increased, the thickness sheet decreased but the strain increased. Particularly, when the fourth pass was completed, the sheet thickness was reduced by 18.3% compared with the previous pass. Moreover, the strain contour shows that the severely affected area was located in the ribbon-rounded-corner regions of 7.2 mm in the middle of the part, especially the protrusion region. Figure 12 shows the effect of the number of forming passes on the thickness uniformity obtained through simulation. It can be observed that the number of forming passes was inversely proportional to the sheet thickness difference and effective plastic strain, i.e., using more forming passes can improve the sheet thickness uniformity.

### 3.3. Tool Moving Direction

Figure 13 shows the simulation results under the conditions of different motion directions. In scheme 1, two counterclockwise passes were used, and in scheme 2, two clockwise passes were used. In scheme 3, the first pass was counterclockwise while the second was clockwise. As shown in Figure 13, the difference between the simulation results under schemes 1 and 2 (same motion direction) was small; however, when comparing the results under schemes 1 and 2 with that under scheme 3 (opposite motion direction), the difference was significant. This indicates that the motion direction, namely same or opposite, had a significant effect on the sheet forming quality. Based on the histogram, scheme 3 shows the best forming effect. The maximum strain of the formed part was 0.837, and the maximum wall thickness difference was 0.542 mm, which was 7.19% lower than that of the reference value. The results demonstrate that with an alternating clockwise and counterclockwise scheme, the simulation of an opposite forming motion trajectory was found to have a positive effect on the sheet uniformity. When the moving direction of the forming tool changed, the material flow direction changed accordingly, particularly for the parts with a large height.

### 3.4. Forming Angle

Table 6 lists the forming angle and simulation results under the three schemes. From the results of the first pass, it can be concluded that as the forming angle increased, the difference between the strain and the maximum thickness increased, and the sheet thickness was more uneven. The results of the second pass indicate that the maximum thickness difference of the formed part in scheme 3 was the smallest, while the deformation between the second and first passes was minimal. From the results obtained under the motion trajectory schemes with three different forming angles, we conclude that the amount of deformation in each pass should be reasonably allocated in a multi-pass TPIF process and the forming angle between different passes should be reduced as much as possible.

### 3.5. Optimal Set of Process Parameters

The simulation result under the conditions of the set of optimal process parameters (the diameter of the tool head was 12 mm, the feed rate was 15,000 mm/min, and the step size was 0.4 mm) acted as the reference. On this basis, the simulation result under the conditions of four passes, opposite motion direction, and a smaller forming angle were obtained. Figure 14 shows the optimal simulation results and the reference. From Figure 14a,b, it can be observed that the effective plastic strain after optimization was lower than before. In Figure 14c, the maximum wall thickness was 0.992 mm, the minimum wall thickness was 0.484 mm, and the maximum thickness difference was 0.508 mm, but in Figure 14d, the maximum wall thickness was 0.996 mm, the minimum wall thickness was 0.414 mm, and the maximum thickness difference was 0.582 mm. Compared with the reference, the optimized group gave better results.

Based on the aforementioned optimal process parameters, 16 parts were processed, as shown in Figure 15. Three formed parts were chosen randomly for measurement using a 3D scanner. Figure 16 shows the coordinate value and thickness of the sheet. Table 7 lists three groups of measurement data, taking the average value as the experimental data *n*. Meanwhile, the corresponding points in the numerical simulation result were selected to obtain the value of the sheet thickness as the simulated data *m*. Table 8 lists the error analysis results. The error δ was defined as:(12)$$\delta =\frac{\left|m-n\right|}{m}\times 100\%$$
where *m* and *n* are the sheet thicknesses from the numerical simulations and physical experiments, respectively. The error δ was influenced by various factors, such as friction, the difference in the smoothness of the surface of the supporting mold or sheet, geometric error related to the device placement, and measurement position error. Although there was a certain degree deviation of the sheet thickness between the experimental and simulated results, the error was within 10%.

## 4. Conclusions

In this study, the influences of process parameters on sheet thickness uniformity using the TPIF process for an AA1060 aluminum alloy were investigated; the main conclusions are summarized as follows.

First, we found that the thickness uniformity of the irregular stepped part was related to the diameter of the tool head, feed rate, and step size based on the orthogonal experiment. From the perspective of importance, the diameter of the tool head had the greatest effect, followed by the feed rate, and the step size has the least effect, where the corresponding optimal values were 12 mm, 15,000 mm/min, and 0.4 mm, respectively. The sheet thickness uniformity improved with the increase of tool head diameter but it significantly decreased with the increase of feed rate and step size.

Second, the relationship equations between the maximum thickness difference and effective plastic strain as a function of the diameter of the tool head, feed rate, or step size were obtained for two-pass TPIF for the target part using a polynomial fit method.

Additionally, in a multi-pass TPIF, more passes and a smaller forming angle improved the wall thickness uniformity. The reduction of the forming angle between each pass was beneficial to the flow of the metal and improvement of the formed part accuracy. By alternating the motion direction of the forming tool between adjacent passes, the wall thickness uniformity was significantly improved.

## Figures and Tables

**Figure 1 materials-13-02634-f001:**
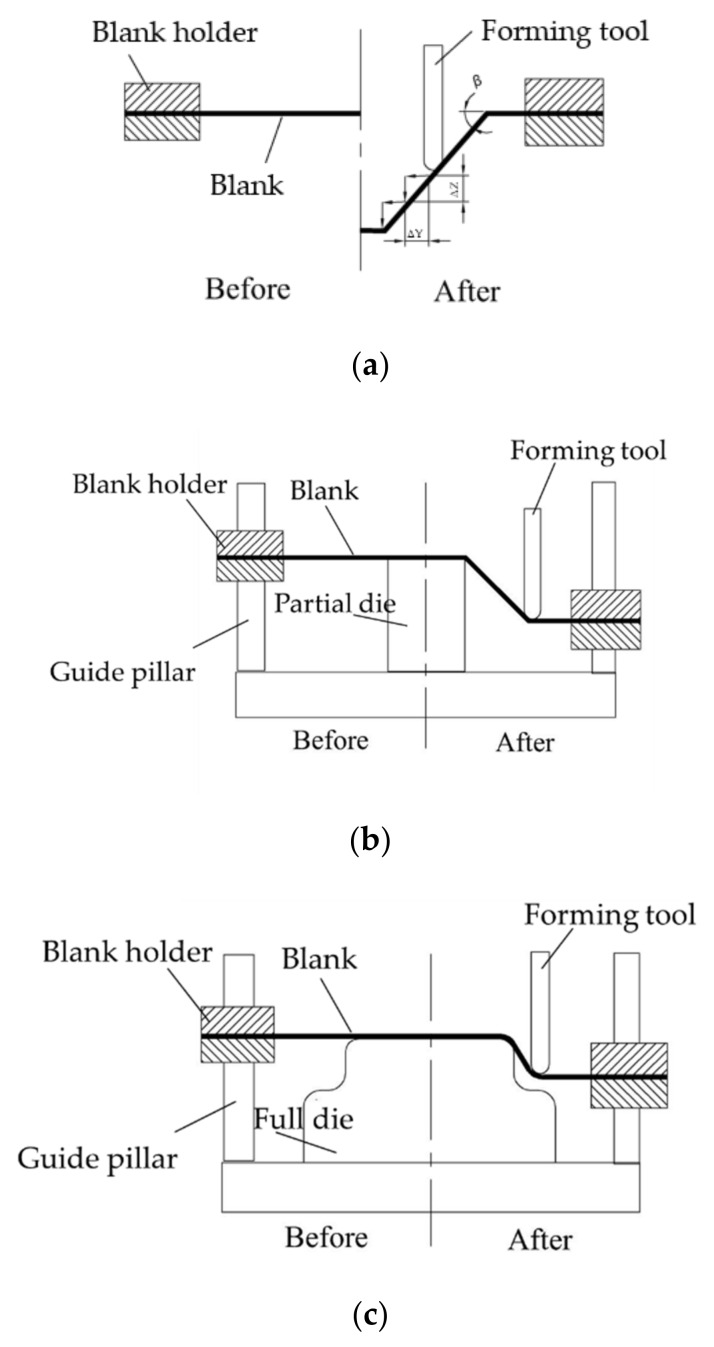
Schematics of incremental sheet forming (ISF): (**a**) single-point incremental forming (SPIF) (or negative incremental forming (NIF)); (**b**) two-point incremental forming (TPIF) (or positive incremental forming (PIF)) with a partial die; (**c**) TPIF (or PIF) with a full die.

**Figure 2 materials-13-02634-f002:**
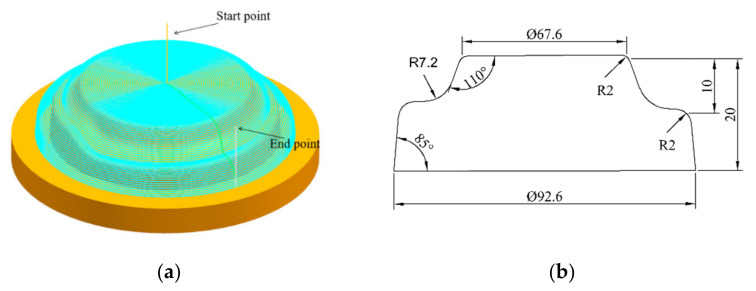
Model of the target part: (**a**) 3D model and single-pass toolpath, and (**b**) dimensions of the part in millimeters.

**Figure 3 materials-13-02634-f003:**
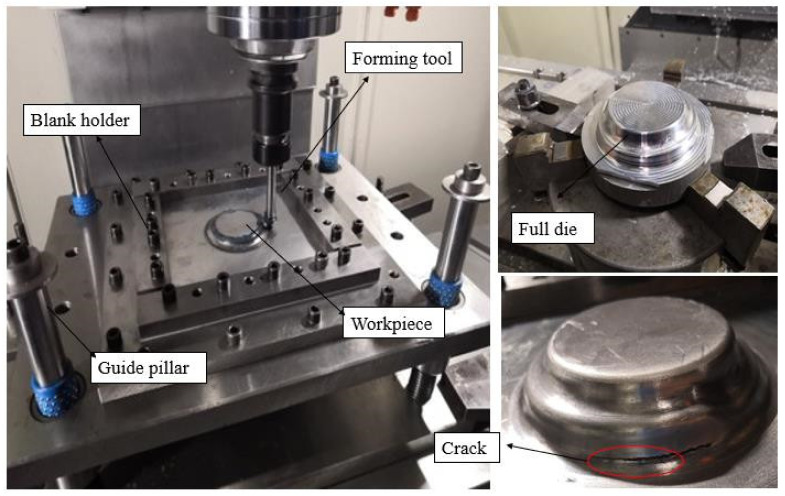
Forming device.

**Figure 4 materials-13-02634-f004:**
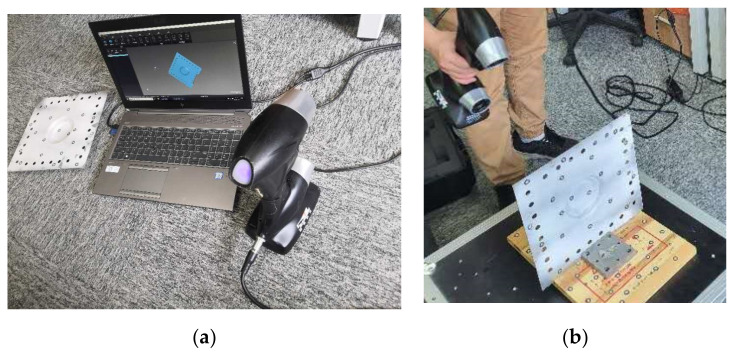
Scanning process and equipment: (**a**) measurement device and (**b**) scanning process.

**Figure 5 materials-13-02634-f005:**
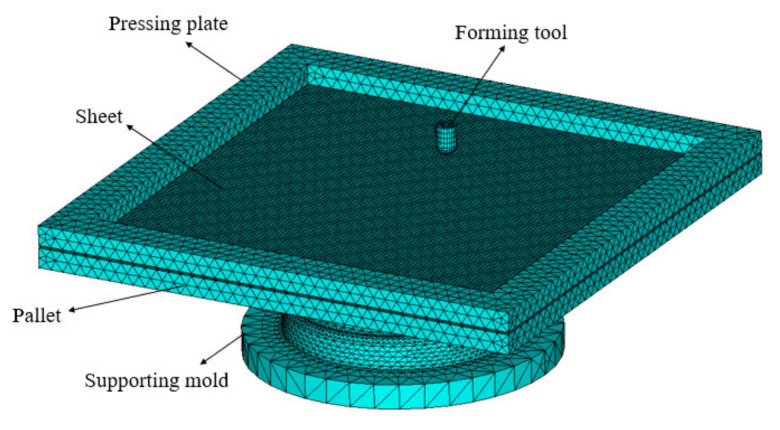
Finite element mesh model.

**Figure 6 materials-13-02634-f006:**
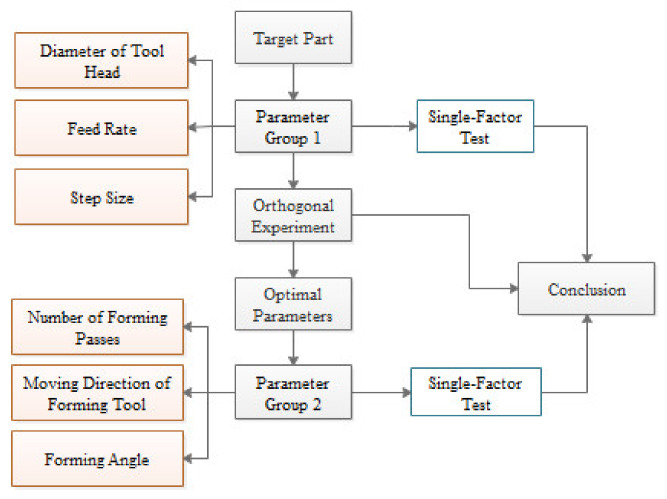
Design process.

**Figure 7 materials-13-02634-f007:**
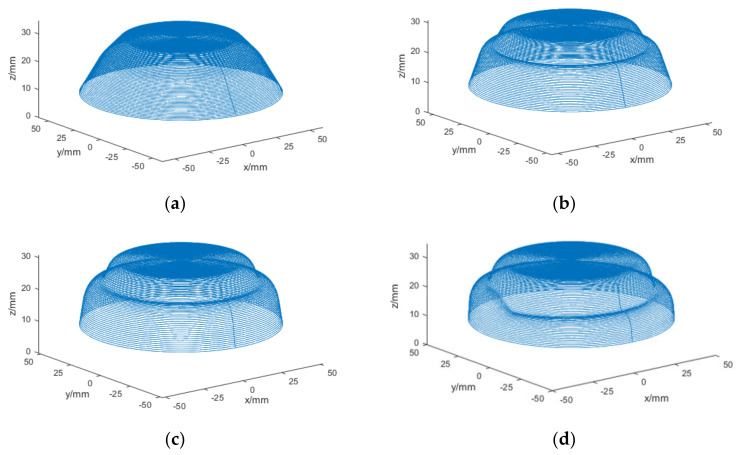
Motion trajectory curves of the irregular stepped part using a four-pass TPIF: (**a**) the first pass, (**b**) the second pass, (**c**) the third pass, and (**d**) the fourth pass.

**Figure 8 materials-13-02634-f008:**
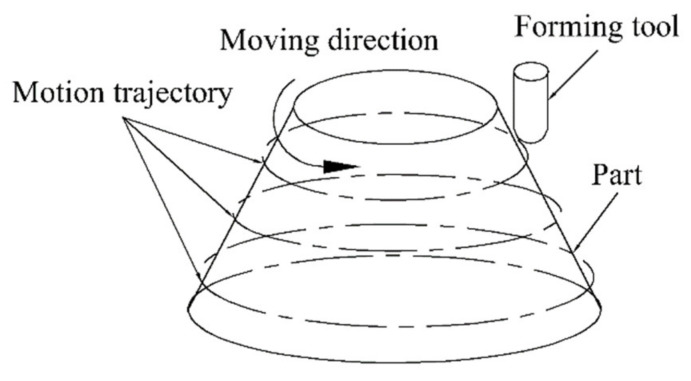
Schematic of the forming trajectory and motion direction.

**Figure 9 materials-13-02634-f009:**
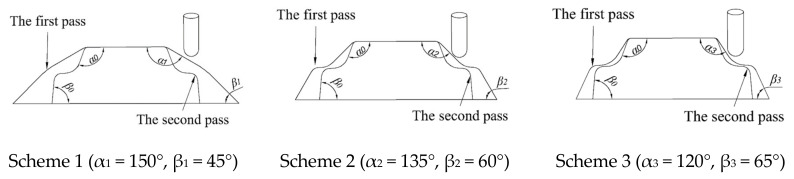
Forming angle schemes.

**Figure 10 materials-13-02634-f010:**
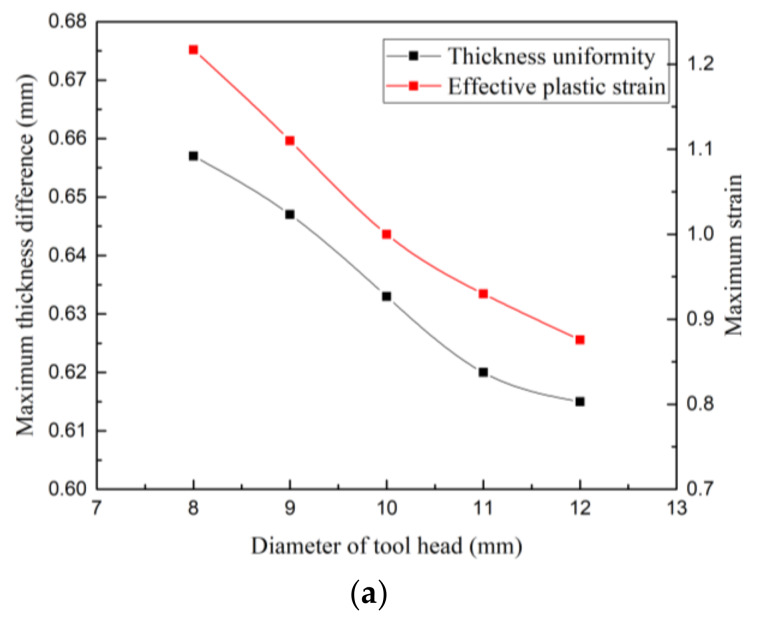
Influence of various process parameters on the maximum thickness difference and maximum strain: (**a**) diameter of the tool head, (**b**) feed rate, and (**c**) step size.

**Figure 11 materials-13-02634-f011:**
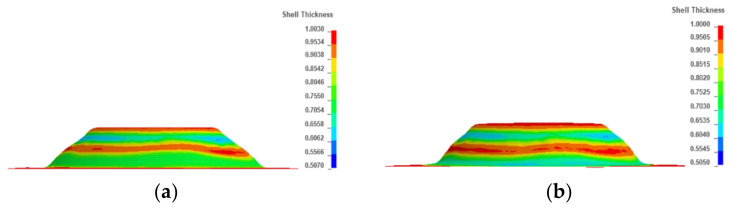
Contours of the thickness and effective plastic strain: (**a**) thickness contour of the first pass, (**b**) thickness contour of the second pass, (**c**) thickness contour of the third pass, (**d**) thickness contour of the fourth pass, (**e**) strain contour of the first pass, (**f**) strain contour of the second pass, (**g**) strain contour of the third pass, and (**h**) strain contour of the fourth pass.

**Figure 12 materials-13-02634-f012:**
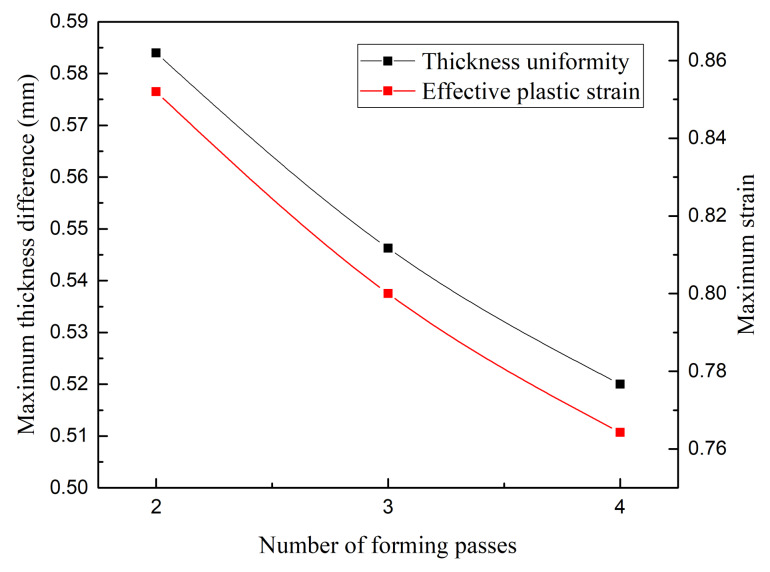
Effect of the number of forming passes.

**Figure 13 materials-13-02634-f013:**
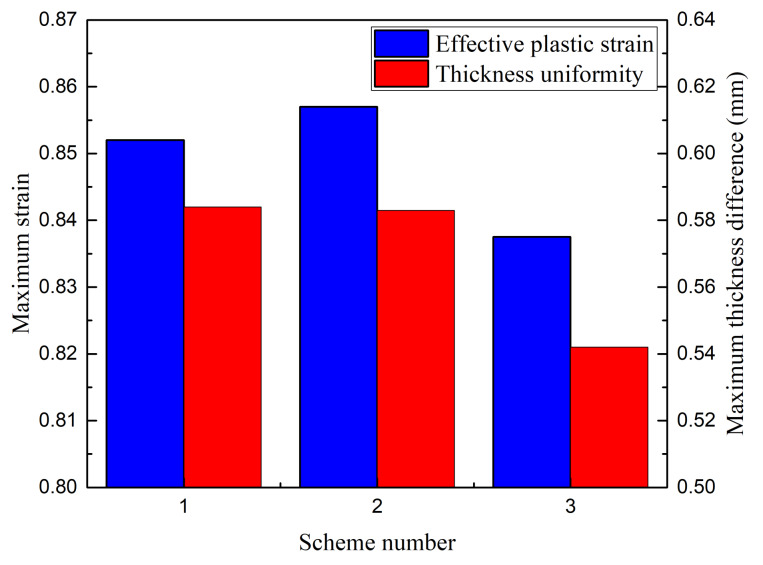
Effect of the number of forming passes.

**Figure 14 materials-13-02634-f014:**
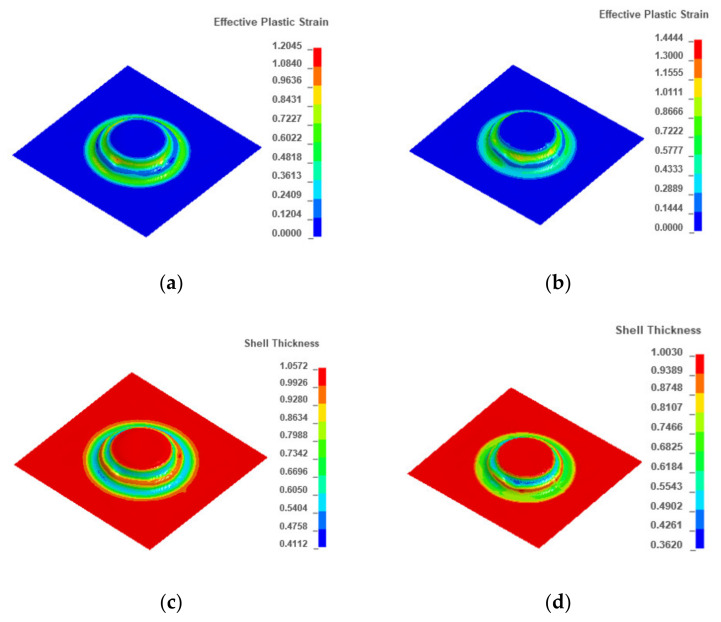
Optimal simulation result and the reference: (**a**) effective strain contour using optimal process parameters, (**b**) effective strain contour of the reference, (**c**) thickness contour using optimal process parameters, and (**d**) thickness contour of the reference.

**Figure 15 materials-13-02634-f015:**
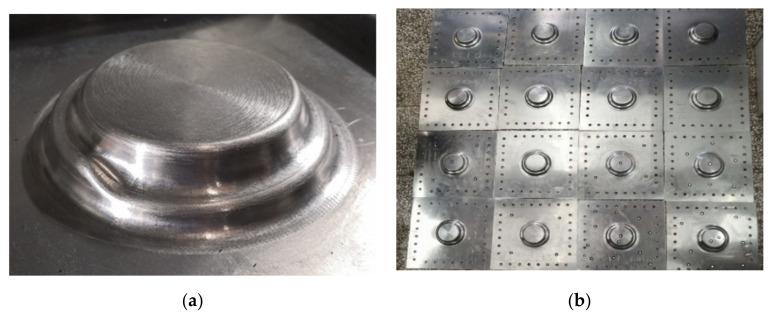
Formed target parts: (**a**) single part, (**b**)total parts.

**Figure 16 materials-13-02634-f016:**
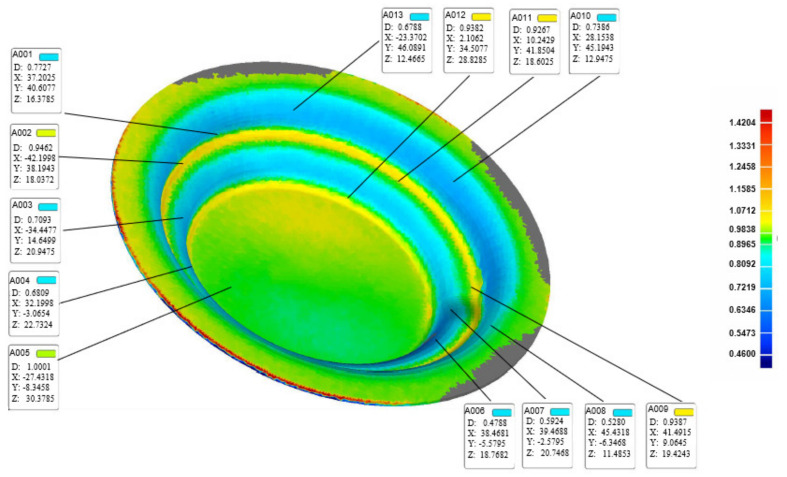
Measurement results.

**Table 1 materials-13-02634-t001:** Material properties.

Parameter	Value (Unit)
Density	2700 (kg/m³)
Modulus of Elasticity	55.94 (GPa)
Poisson’s Ratio	0.324
Strength Coefficient	197.75 (MPa)
Strain Hardening Exponent	0.12
Barlat Exponent (*m*)	8
*R* _0_	0.686
*R* _45_	1.307
*R* _90_	0.641

**Table 2 materials-13-02634-t002:** Experimental factors and levels.

	A	B	C
Level	Diameter of Tool Head (mm)	Feed Rate (mm/min)	Step Size (mm)
1	8	15,000	0.4
2	10	16,500	0.6
3	12	18,000	0.8

**Table 3 materials-13-02634-t003:** Plan of orthogonal experiment and result.

No.	A	B	C	Effective Strain	Thickness Difference (mm)
1	1	1	1	1.000	0.638
2	1	2	2	1.120	0.660
3	1	3	3	1.323	0.674
4	2	1	3	0.971	0.640
5	2	2	1	1.055	0.634
6	2	3	2	1.120	0.642
7	3	1	2	0.870	0.605
8	3	2	3	0.895	0.614
9	3	3	1	0.924	0.625

**Table 4 materials-13-02634-t004:** Results of the range analysis of the maximum strain.

Level	A	B	C
1	1.148	0.947	0.993
2	1.049	1.023	1.037
3	0.896	1.122	1.063
D	0.251	0.175	0.07

**Table 5 materials-13-02634-t005:** Results of the range analysis of the maximum thickness difference (mm).

Level	A	B	C
1	0.657	0.628	0.632
2	0.639	0.636	0.636
3	0.615	0.647	0.643
D	0.043	0.019	0.010

**Table 6 materials-13-02634-t006:** Simulation results based on different forming angles.

	Forming Angle		Effective Strain		Thickness Difference	
Scheme	Upper	Lower	First Pass	Second Pass	First Pass	Second Pass
Scheme 1	30°	45°	0.594	0.852	0.303	0.582
Scheme 2	45°	60°	0.650	1.160	0.387	0.693
Scheme 3	60°	65°	0.680	0.811	0.450	0.511

**Table 7 materials-13-02634-t007:** Measurement data.

Point	Reference Positions			Measure Results		
	X	Y	Z	1	2	3
A001	37.2025	40.6077	16.3785	0.7727	0.7989	0.7738
A002	−42.1998	38.1943	18.0372	0.9462	0.9233	0.9587
A003	−34.4477	14.6499	20.9475	0.7093	0.7437	0.7366
A004	−32.1998	−3.0654	22.7324	0.6809	0.6784	0.6358
A005	−27.4318	−8.3458	30.3785	1.0001	0.9985	0.9803
A006	38.4681	−5.5795	18.7682	0.4788	0.4922	0.4603
A007	39.4688	−2.5795	20.7468	0.5924	0.5839	0.5883
A008	45.4318	−6.3468	11.4853	0.528	0.4735	0.5
A009	41.4915	9.0645	19.4243	0.9387	0.9524	0.9584
A010	28.1538	45.1943	12.9475	0.7386	0.7283	0.7589
A011	10.2429	41.8504	18.6025	0.9267	0.939	0.9394
A012	2.1062	34.5077	28.8285	0.9382	0.9653	0.9782
A013	−23.3702	46.0891	12.4665	0.6788	0.66	0.6642

**Table 8 materials-13-02634-t008:** Error analysis.

Point	Simulation (mm)	Experimental (mm)	Error (%)
A001	0.7498	0.7818	4.2678
A002	0.9567	0.9427	1.4599
A003	0.7455	0.7299	2.0970
A004	0.6820	0.6650	2.4878
A005	0.9980	0.9930	0.5043
A006	0.4922	0.4771	3.0679
A007	0.5700	0.5882	3.1930
A008	0.5353	0.5005	6.5010
A009	0.9762	0.9498	2.7009
A010	0.7158	0.7419	3.6509
A011	0.9460	0.935	1.1593
A012	0.9384	0.9606	2.3622
A013	0.6704	0.6677	0.4077

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
