# Peer review of "Effects of Process Parameters on the Thickness Uniformity in Two-Point Incremental Forming (TPIF) with a Positive Die for an Irregular Stepped Part"

_materials, 2020, doi:10.3390/ma13112634_

Round 1

Reviewer 1 Report

The originality of the presented paper resides mainly in the output chose (thickness uniformity, usually refers to thinning in incremental sheet forming field), the used approach that includes numerical simulations, optimization of the tool path and experimental validation, and that it is a two-point experimental work but with partial die.

The main insurmountable drawbacks of the work are:

  1. The most important, the authors are talking about single point incremental forming when they are actually adopting a two-point incremental set-up, but with partial die. They need to revise the literature review also for correctly use the name of the process parameters (step-down instead of their spacing) and some other inconsistencies that appear in the paper.

I propose just some references that should revise, for example:

  • For Incremental Forming classification: (Allwood et al., 2010). Allwood, J.M., Braun, D., Music, O., 2010. The effect of partially cut-out blanks on geometric accuracy in incremental sheet forming. Journal of Materials Processing Technology 210, 1501– doi:10.1016/j.jmatprotec.2010.04.008

Jackson and Allwood (2009) concluded that the deformation mechanism is intrinsically different for SPIF and TPIF. Therefore the two processes should be correctly distinguished.

  • One of the few works where TPIF is used although with polymers. “Customized cranial implant manufactured by Incremental Sheet Forming using a biocompatible polymer”, in Rapid Prototyping Journal on May 2017, doi: 10.1108/RPJ-06-2016-0089 (Bagudanch et al., 2018). It demonstrates the feasibility of ISF considering the SPIF and TPIF process variants to produce customized cranial implants using a biocompatible polymer (UHMWPE), ensuring an appropriate geometric accuracy and low cost.
  • Because if the stepped geometry and output studied: Manco, L. Filice, G. Ambrogio, Analysis of the thickness distribution varying tool trajectory in single-point incremental forming, Proc. Inst. Mech. Eng. Part B J. Eng. Manuf. 225 (2011) 348–356, doi:10.1177/09544054JEM1958
  1. The structure of the work. The order in which the sections are introduced. The reviewer recommend that the authors revise the classical and well-known 2005 Ashby paper paper, “How to write a paper”.

Some examples to consider:

  • Material should be indicated in the introduction, even in the abstract. The main papers they used as references should be explicitly referred to, because in section
  • Experimental Set-up description is missing, it should be introduced before numerical simulation. Even if it is a final validation step, a scheme of the methodology followed should be included.
  • Feed rate range values are low, the units are not the usual. Why did the authors choose these values?
  • The stepped geometry description should be introduced previously, it is not introduced until section 2.4.3. Why the angles of the stepped geometry have not been considered as parameters to be studied? Did the authors make some experiments to determine limit angle or they used published works for 1060 aluminium? They should provide the references.
  • In section 2.4, line 139, what is the forming angle? Can the experimental set-up move the axis of the tool?
  • Suddenly, the outputs are introduced in table 3, but ii is not explained how they are measures, or which are the equipment used?
  1. Effects of process parameters is not based on any statistical analysis.
  2. Optimization has a lot of importance in the title but are the authors sure they are making an optimization procedure or just choosing the best parameters??

According to the above main comments, the paper should be re-framed. The authors should improve the presentation of the results and their discussion and, finally, they must write a new conclusion section accordingly.

Reviewer 2 Report

The title of this paper is “Optimization of motion trajectory on single point incremental positive forming for spiral stepped parts” and in my opinion the title does not reflect the content of the paper. The authors mention in the abstract (line 15) and in the chapter 2.3 (line 127)  that  the influence factors taken into consideration are besides feed rates, the punch diameter and ”spacing”. The punch diameter is a geometrical (technological) factor not a factor that determines the motion trajectory. And please explain what it means ”spacing” because it is not clear in the paper and, in my opinion, it is not a factor of influence of the motion trajectory. In the chapter 2.4  ”Motion trajectory design” (line 136) the authors mentioned that ”the design of motion trajectory scheme is based on three aspects: the number of forming passes, direction of movement of the forming tool, and forming angle”. I think that the number of forming passes, and forming angle are the most important for SPIF while the direction of movement of the forming tool (clockwise or counterclockwise!!) has a smaller importance. For me it is not clearly defined the aim of the paper. I understand that, in order to decrease the number of experiments, they split the influence factors but, in my opinion, they could even eliminate the punch diameter and the direction of movement of the forming tool. Of course, the punch diameter has an outstanding importance on the thickness uniformity, but it is not a part of motion trajectory! Otherwise the authors could keep punch diameter and change the title of the paper.

Line72 ”The SPIF technology has mainly been applied to form simple axisymmetric parts [18,19,20,21], including conical, square conical, and straight wall parts, with little attention to complex parts.” Please detail why you include these references in the paper. And why you put these references on the same phrase? Only to exemplify which kind of geometries could be produced by SPIF? In the first chapter (Introduction) which is in fact the ”state of the art” in this filed, I read that there are only few references that refer to the optimization of the trajectory (3 references, detailed from line 53 to line 60). I know that are many other papers that study the optimization of the trajectory. I could recommend you few of them:

Behera, A.K., Verbert, J. Lauwers, B., Duflou, J.R. Tool path compensation strategies for single point incremental sheet forming using multivariate adaptive regression splines, Computer-Aided Design, 2013, 45, 3, 575-590

Blaga, A., Oleksik, V. A Study on the Influence of the Forming Strategy on the Main Strains, Thickness Reduction, and Forces in a Single Point Incremental Forming Process, Advances in Materials Science and Engineering, 2013, 382635

Hu, Z., Jin, J., Jinlan, B. Research on the forming direction optimization for the uniformity of the sheet part thickness in the CNC incremental forming, The International Journal of Advanced Manufacturing Technology, 2017, 93, 2547–2559

Figure 2. I do not understand why the authors have meshed the entire geometry of the supporting mold. They could mesh only the active geometry that is in contact with the sheet. Increasing the number of the elements conducts to increasing the processing time even though they defined the supporting mold as rigid body.

Line 121, 122 The authors wrote ”The contacts between the sheet metal and the supporting mold, the upper and lower pressing plates, and the tools are considered surface-to-surface with a friction coefficient of 0.2.” They did not mention anything about the lubrication of the process. I could not say if the value of the friction coefficient is correct. Usually, in Ls-Dyna we choose for lubricated contact for forming process 0.08 to 0.1. Also, I do not understand why they used the surface-to-surface contact and they did not use the forming one-way surface to surface contact which is better for the forming simulation. I recommend also to use 7 or 9 integration points for other simulations, in order to increase the precision of the analysis. But this is not mandatory.

Line 123, 124. The authors wrote ”In the forming process, the rotational degree of freedom of the forming tool is restricted, thus the forming tool moves only along linear direction”. I understand that they wrote that the punch and of course the tools are not allowed to rotate but it is not correct to say that the forming tool moves only along linear direction. Obviously, the tools follow combined (linear and nonlinear) trajectories.

Line 158-165. Please specify the 6 values of angle alpha and beta.

Line 178. The authors wrote that ”A3B1C1 represents the optimal set of parameters: the diameter of the tool head is 6 mm, the feed rate is 250 mm/s, and the spacing is 0.4 mm”. But in the abstract chapter they wrote ”optimal values are 12 mm, 250 mm/s and 0.4 mm” and on the chapter 3.2.1. Diameter of Tool Head (line 183) they mentioned that ”we can conclude that a larger tool diameter leads to a smaller maximum thickness difference and a more uniform deformation” (line 189). On the conclusion chapter also, they mentioned that ”The optimal process parameter combination is the diameter of the forming tool head of 12 mm, the feed rate of 250 mm/s, and the spacing of 0.6 mm” (line 312). I suppose that it is only a mistyping on line 178.

Line 181. Why in the table 4 they introduce A,B and C instead of punch diameter, feed rates and spacing? Also, they did present in table 4 the results for maximum thickness difference but they did not present the results for plastic strain. They refer to the plastic strain in chapters: 3.2.1. Diameter of Tool Head, 3.2.2. Feed Rate, 3.2.3. Spacing but they did not present the results. I suggest introducing another table with the plastic strain results.

Lines 225-226. Authors wrote: ”the strain contour shows that the severely affected areas on the sheet are the rounded corners of the stepped part”. Which rounded corners? This is not a spiral stepped shape part?

Figure 8. In the legend, please replace ”Thickness” with ”Thickness uniformity”. It is very important to not confuse these terms.

Line 267. Optimization. I consider that the entire subchapter must be reevaluated! They consider the maximum difference (the uniformity of the thickness) between maximum thickness (1.0 mm) and minimum thickness after the SPIF process. You cannot compare the maximum thickness, which is, practically, the initial thickness of the sheet, because there are many areas on the part which are not affected by the forming process! They must calculate the difference between areas that are affected by the forming process!

Line 288. The authors present the experimental layout and an 3D laser scanner used for measuring the parts. They did not mention the name of this laser, they did not mention if they measured the parts at the end of the process or during the forming process. They did not mention what they measured. I suppose that they measured the thickness of the part. They only present a photo of the scanner and an arrow on the photo of the experimental layout.

Line 292. They must present a figure taken from the measurement process not only a table with 9 points and the corresponding errors (table 6). They did not mention how many measurements they did. They did not mention how they calculated the error. They calculated the error like a difference between simulation and experimental result?

Line 303. The conclusion chapter.

Many of the conclusions are well known. Increasing of the diameter of the tool head eliminates the strain concentrator because the strains are distributed on a bigger surface. Increasing the number of the steps conducts in improving the thickness uniformity but also in increasing of the thinning. Also, alternating the motion direction of the forming tool it is well known that it is possible to improve the thickness uniformity.

Otherwise, on the entire paper, the authors did not mention anything about the ”sine law”, a law that is unanimously accepted for determining the thickness on the parts manufactured by SPIF. So, based on this law (which tells that the thickness of the part is directly proportional with the sinus of the wall angle) it is obvious that ”A smaller forming angle is advised” (line 320).

Another mistyping (I guess) is present on the conclusion chapter, referring to the optimal spacing: ”The optimal process parameter combination is the diameter of the forming tool head of 12 mm, the feed rate of 250 mm/s, and the spacing of 0.6 mm” (line 312). In the chapter 3.2.3. Spacing and in the abstract, they declare that the optimal value for the spacing is 0.4 mm!!!

As a conclusion, I consider that the paper must be significantly improved before publishing. The paper cannot be published in this form.

Reviewer 3 Report

In the paper the optimization of SPIF is carried out considering many parametersas tool path, tool diameter, feed rate……

Also if some of the conclusions about some of these parameters are not innovative, the global analysis could be interesting if the comparison with the previous study is considered.

In particular the references reported in introduction are totally inadequate.

In the paper the authors consider the toolpath strategy as clokwise and counterclockwise, but in introduction any paper treating this strategy is cited.

The authors cited in the introduction papers about the use of temperature in order to increase the formability, but they don’t cite paper in which the influence of tool diameter and feed rate were studied.

Finally, the discussion should be based on the comparison with the results of previous research that are not cited in introduction.

Reviewer 4 Report

The paper titled “Optimization of motion trajectory on single point 2 incremental positive forming for spiral stepped parts” gives an interesting deepening about the single point positive incremental forming, leading the final results to the process optimization based on a preliminary step based on the numerical simulation and the subsequent validation via experimental trials.

Despite the topic is of great interest, the paper in its present form is not suitable to be directly published. In the reviewer’s opinion, the paper could be reconsidered only after major revision, regarding not only a full check on the manuscript language but also the improvement of some of the technical contents. In particular, the following points need to be discussed more deeply:

  • A more detailed description of the target component is provided: the depth information is missing (both the total value and those regarding each stepped section); the target wall angles are not explicitly mentioned with the exception of the one equal to 85° which is declared only in the section 2.4.3. In addition, according to Figure 2a, the final component is characterized by a characteristic feature (a sort of protrusion on the left side) which is not discussed and which seems to have a strong influence on the capability of the process to reproduce the target geometry (in particular on the number of passes).
  • According to what reported on line 116, the authors set the value of the m exponent (Barlat yield function) at 6; it is reported in literature (for example, the investigation by Hosford in 1980) that the m exponent is usually set at 8 in the case of FCC materials. The authors should motivate their choice.
  • Are data in Table 1 taken from literature? If so, a reference should be added. In addition, (i) the unit of the Young’s modulus should be corrected in GPa, (i) the “Hard Coefficient” (which should be more correctly defined as “Strength Coefficient”) is missing its unit and (iii) the “Width ratio” is somehow confusing and it would be preferable to replace with the label R (as reported in the manuscript).
  • The authors adopted the friction coefficient equal to 0.2 (line 121-122): is that value taken from literature or experimentally measured?
  • The authors should discuss better their choice not to consider the tool rotation (line 123-124): was it due to the negligible influence of the spindle speed on the outcome of the process or was it intentionally omitted in the investigation?
  • According to the contents of paragraphs 2.3 and 2.4, the adopted methodology was based on a preliminary investigation of the input variables effect (diameter of the tool head, feed rate and spacing) on the outcome of the process (strain and thickness uniformity); from this preliminary investigation, an optimal combination was defined as “reference”. Then, the influence of the number of passes, the direction of the tool movement and the forming angle were investigated. If so: (i) Effective strain and Thickness difference are not fully defined: is the effective strain calculated in the most strained point of the wall? Is the thickness difference calculated as the difference between the most and less strained point along the wall or between the most strained point and the undeformed blank thickness? (ii) Is the “L0 (34)” definition correct? It seems that the orthogonal array should be L9 (33) being three input variables varied on 3 levels each. In addition, referring to the operative conditions in Table 3 as “experiment” is somehow confusing, giving the idea of experimental test rather than numerical runs. (iii) The forming conditions in Table 2 regard to a 2-passes forming strategy, but there is no explicit mention about the direction of the tool movement and/or the forming angles. (iv) According to Figure 3, only a 4-passes strategy is able to fully reproduce the target geometry: in the other cases, none of the strategies is able to reproduce the feature discussed in the first point (highlighted by the red circle), which leads to the following point: why do the author consider the other 3 strategies (1, 2 and 3 passes) if none of them would have been able to fully get the final component? (v) The paragraphs 2.4.2 and 2.4.3 refer to the influence of the moving direction of the tool and the forming angle, but all the reported results are relative to a 2-passes strategy: what about the influence of those parameter on the 3 and 4-passes strategies? (vi) Do the authors consider the possibility to carry out a single step investigation, thus incorporating the number of passes, the moving direction of the tool and the forming angles as additional variable inputs (beside the tool head diameter, the feed rate and the spacing)? What is the reason of splitting the investigation in a two-steps procedure?
  • Paragraph 3.1 contains the numerical results (line 177-180) but the following points need to be better clarified: (i) how do the authors define the A3B1C1 as the optimal solution being that particular combination not present in Table 2?; (ii) the authors state that the A3B1C1 is characterized by a tool diameter of 6 mm, but probably they intend 12 mm (6 mm is not even mentioned in the investigated levels of the orthogonal table); (iii) why is the A3B1C1 compared with the condition #3 (A1B3C3)?; (iv) Table 4 reports the results of the range analysis also in terms of the parameter “R” which not only is not defined, but it may also be confused with the anisotropy coefficient.
  • Paragraphs 3.2.1, 3.2.2 and 3.2.3 report the effect of the process parameters (diameter of the tool head, feed rate and spacing) on the defined output variable, but it is not clear the reason why the authors need to investigate also the intermediate values (9 and 11 for the tool diameter, 270 and 310 for the feed rate, 0.5 and 0.7 for the spacing), remarkably increasing the number of numerical runs and thus partially nullifying the advantages of the orthogonal table.
  • How do the authors define the optimized process conditions in the paragraph 3.4 according to the results of paragraphs 3.3.2 and 3.3.3 (which were still based on a 2-passes strategy)? What about the forming angles and the alternate direction of the tool (it is only reported that the moving direction was alternate)?
  • Being the paragraph 3.5.1 a description of the adopted facility, it should be moved in the section regarding the description of the methodology.
  • The paragraph 3.5.2 regards the experimental verification of the numerical predictions: what about the replicability of the process? How many replications were done of the optimal conditions? What about the geometrical correspondence of the final component with the target geometry?

Round 2

Reviewer 1 Report

The improvement of the paper is considerable. In reviewer's opinion there are some minors things to solve (listed below), and several important issues.

  • The first, in line 191 the authors refers to NIF (supposed Negative Incremental Forming) and to Figure 3 when it is supposed PIF (Positive Incremental Forming). In reviewer’s opinion, the authors should refer their work as TPIF (Two point Incremental Forming) with positive die. In that way, they will fit with the classic nomenclature from the literature.
  • The second main issue is related to optimization procedure that the authors claim in the title and in section 2.2, for example. This was a previous question that it is unsolved, optimization has a lot of importance in the title but are the authors sure they are making an optimization procedure or just choosing the best parameters?? When one read about optimization is thinking in another kind of work. In my opinion, the work worth but all the optimization importance is confusing. The authors should clarify or to modify the approach. Section 2.2 is not clear, neither the approach not the methodology of the supposed optimization. 
  • A methodology section, that explain the overall approach of the work will help for the general purpose and understanding of the chole paper. Because suddenly in Results section, it appears range analysis.

  • According to the previous comment, results sections must be improved.In table 2 refers to 3 levels, for example for tool diameter 8,10 and 12mm. A table 3 with an experimental plan but then, in the results appears also tool dimeter of 9 and 11. The same happens for the rest of the parameters studied.

  • In consequence conclusion should be aligned with the two previous comments.

Minor comments:

General English revision should be done.

What does it mean “spiral stepped part”(line 130)? That the stepped part has really the spiral (not noticed in the Figure 2) or that the tool path is spiral (which does not correspond with the explanation in paragraph in section 2.1.1  lines 144-45.

Letters missing at the end of line 150

Does Figure 4 b corresponds to the real PIF part formed by the authors? It cannot be easily appreciated.

Which is the material of the full die? How it has been manufactured?

What does it mean 195 line “(…) Meanwhile, in order to better reflect the relationship between the response and the input variables, the intermediate value is increased and investigated.”?

Are the authors sure that Feed rate provided valued, are they correct? can the machine really provide the feed rate values?

What the authors refers as “Forming passes” in section 2.2.2 should be move up close to section 1.1.1. Due to they refer to it previous the explanation of this section. How the authors obtain the toolpaths?

How many parts have been manufactured? How many repetitions have you done for the experimental plan?

How strain measurement is done? For what reason are considered and carried out?

Reviewer 2 Report

Response 1

Based on the content of the article and your advices, the title has been modified as

optimization of process parameters on thickness uniformity in positive incremental

forming (PIF) with full die for spiral stepped part’.

ACCEPTED

Response 2:

In addition, we replaced the name of the process parameters “spacing “with “step size”,

which is denoted by ΔZ illustrated in Figure 1 (a).

ACCEPTED

Response 3:

The title of this article has been changed, so we revised this part accordingly and deleted and

added many literatures in introduction section.

ACCEPTED

Response 4:

The Finite element mesh model as shown in Figure 5, the sheet metal will contact with the

supporting mold gradually, the boundary and contact conditions need to be set, so we have

meshed the entire geometry and set the friction coefficient. Mesh is employed for the sheet

metal with a mesh size of 1.5 mm. to save calculating time, a bigger element size was taken

for blank holder and supporting mold.

PARTIALLY ACCEPTED

They could have meshed only the active geometry and the blank. I refer to the base of the supporting mould! But it is ok, that only increases the time of the analysis.

Response 5:

Friction coefficient 0.2 is cite form literature [34]. In LS-DYNA, master-slave surface method is

used for automatic contact (surface-to-surface contact). Face to face contact is characterized

by the penetration of one face into another. According to the characteristics of the contact

between the forming tool and the sheet in SPIF, that is, the sheet will become thinner and

even break during the forming process, the forming tool will penetrate the sheet metal, so the

face-to-face contact is adopted.

ACCEPTED

It was only a recommendation for future analysis.

Response 6:

We really didn't express it clearly, so we modified the expression as ‘ In the forming process,

the forming tool drives the sheet metal movement according to the preset numerical control

program without rotation and only linearly move in X and Y in each layer; The pressing plate

and pallet only move downward along the guide pillar when the forming tool press down.

the supporting mold is stationary, so the total six degree of freedom was restricted.’

ACCEPTED

Response 7:

We specified the angle alpha and beta and show it in Fig.6

ACCEPTED

Response 7:

That is a mistyping, we have correct it.

ACCEPTED

Response 8:

A, B, C was introduced just to simplify the table. According to your advice, we added the

Table 4 and listed the plastic strain results

ACCEPTED

Response 9:

The severely affected areas is located in the ribbon rounded corners regions of 2mm on

the top and in the middle of the part, and it can be seen from Figure 2 (b), the protrusion

section of the spiral stepped part is included in these regions certainly. we have revised

the description

ACCEPTED

Please specifiy this in the paper

Response 10:

”Thickness” has been revised as ”Thickness uniformity”, meanwhile, ”Strain” has been

modified with ”Effective plastic strain”.

ACCEPTED

Line 267. Optimization. I consider that the entire subchapter must be reevaluated! They

consider the maximum difference (the uniformity of the thickness) between maximum

thickness (1.0 mm) and minimum thickness after the SPIF process. You cannot compare the

maximum thickness, which is, practically, the initial thickness of the sheet, because there are

many areas on the part which are not affected by the forming process! They must calculate

the difference between areas that are affected by the forming process!

Response 11:

The maximum and minimum thickness are obtained from the simulation, and the experiment

is used to verify the accuracy of the simulation. The difference of the maximum thickness is

very small, and the thickness uniformity of the sheet mainly depends on the minimum

thickness.

NOT ACCEPTED

As I wrote on my first review, the maximum thickness, is, practically, the initial thickness of the sheet. Of course, that „The difference of the maximum thickness is very small”

Line 288. The authors present the experimental layout and an 3D laser scanner used for

measuring the parts. They did not mention the name of this laser, they did not mention if

they measured the parts at the end of the process or during the forming process. They did not

mention what they measured. I suppose that they measured the thickness of the part. They

only present a photo of the scanner and an arrow on the photo of the experimental layout.

Response 12:

In this part, we add the relative introduce of forming equipment and measurement

Method (Figure 4), and the scanning result is shown in Figure 14 (a).

NOT ACCEPTED In figure 14, a is not the part that the authors present in the rest of the paper. Also, the authors do not present a detailed table with the results of the experimental measurements or a screen capture from the software used for measurement.

Response 13:

Figure 14 shows the result of measurement. A total of three measurements were carried

out, and the average of results was taken as the experimental data n, the error can be

defined as

? =

|? − ?|

?

× 100%

Where m and n are the thickness of simulated and experimental.

ACCEPTED

Response 14:

The conclusion has been revised according the result and discussion of the investigation.

Please check it in the revised manuscript in details.

ACCEPTED

Response 15:

That is a mistyping and we have revised it.

ACCEPTED

Response 16:

The paper has been re-framed and revised.

ACCEPTED

Line 409 I do not understand the expression „the accumulation of materials will be eliminated to a certain extent”

In the paper title, the authors used PIF (positive incremental forming) and in the conclusion chapter and the rest of the paper they used NIF (negative incremental forming). I do not understand why!!!

Reviewer 3 Report

NO COMMENT

Author Response

We deeply appreciate your consideration of our manuscript, if you have any queries, please don’t hesitate to contact me.
Thank you and best regards.

Reviewer 4 Report

The authors have been extensively replied to the questions and clarifications pointed out by the Reviewer.

Author Response

We have revised the expression and style of the manuscript, please check the revised manuscript in details.